# Co-Transplantation of Barcoded Lymphoid-Primed Multipotent (LMPP) and Common Lymphocyte (CLP) Progenitors Reveals a Major Contribution of LMPP to the Lymphoid Lineage

**DOI:** 10.3390/ijms24054368

**Published:** 2023-02-22

**Authors:** Victoria Michaels, Smahane Chalabi, Agnes Legrand, Julie Renard, Emmanuel Tejerina, Marina Daouya, Sylvie Fabrega, Jérôme Megret, Robert Olaso, Anne Boland, Jean-François Deleuze, Christophe Battail, Diana Tronik-Le Roux, Sophie Ezine

**Affiliations:** 1Université Paris Cité, CNRS, INSERM, Institut Necker Enfants Malades-INEM, 75015 Paris, France; 2Université Paris-Saclay, CEA, Centre National de Recherche en Génomique Humaine (CNRGH), 91057 Evry, France; 3Université Paris-Saclay, INRAE, AgroParisTech, GABI, 78350 Jouy-en-Josas, France; 4Atomic Energy and Alternative Energies Agency (CEA), Department of Research in Hemato-Immunology (SRHI), Saint-Louis Hospital, 75010 Paris, France; 5Université Paris Cité, IRSL, HIPI-UMRS 976, 75010 Paris, France; 6SFR Necker—US24/UAR 3633/—Structure Fédérative de Recherche Necker Plateformes Vecteurs Viraux et Transfert de Gènes et Cytométrie, Faculté de Médecine de Necker, 75015 Paris, France; 7Université Grenoble Alpes, IRIG, Laboratoire Biosciences et Bioingénierie pour la Santé, UA 13 INSERM-CEA-UGA, 38000 Grenoble, France

**Keywords:** lymphoid progenitors, cell labeling, genome barcoding, thymus, bone marrow graft, lentivirus, differentiation, cytometry, high-throughput DNA-sequencing

## Abstract

T cells have the potential to maintain immunological memory and self-tolerance by recognizing antigens from pathogens or tumors. In pathological situations, failure to generate de novo T cells causes immunodeficiency resulting in acute infections and complications. Hematopoietic stem cells (HSC) transplantation constitutes a valuable option to restore proper immune function. However, delayed T cell reconstitution is observed compared to other lineages. To overcome this difficulty, we developed a new approach to identify populations with efficient lymphoid reconstitution properties. To this end, we use a DNA barcoding strategy based on the insertion into a cell chromosome of a lentivirus (LV) carrying a non-coding DNA fragment named barcode (BC). These will segregate through cell divisions and be present in cells’ progeny. The remarkable characteristic of the method is that different cell types can be tracked simultaneously in the same mouse. Thus, we in vivo barcoded LMPP and CLP progenitors to test their ability to reconstitute the lymphoid lineage. Barcoded progenitors were co-grafted in immuno-compromised mice and their fate analyzed by evaluating the BC composition in transplanted mice. The results highlight the predominant role of LMPP progenitors for lymphoid generation and reveal valuable novel insights to be reconsidered in clinical transplantation assays.

## 1. Introduction

The development of T cells requires the establishment of complex transcriptional networks, which gradually and hierarchically limit and exclude alternative fates. Failure to generate new T cells results in acute infections and complications caused by immunodeficiency. In pathological states, hematopoietic stem cells (HSC) transplantation would constitute an option to restore proper immune function and accomplish the rescue of the host.

HSC biology has been extensively analyzed, and new insights into their functions, proliferation and lineage specification allow for the improvement of transplantation outcomes. In the adult mouse, all the functional HSC activity is found within the Lin^-^Sca1^+^c-Kit^+^ (LSK) compartment, which constitutes approximately 0.0014% of total bone marrow (BM) cells. HSC have the capacity to self-renew to maintain their own cell pool and can differentiate into downstream progenitors, identified in mice as the lymphoid-primed multipotent progenitor (LMPP) compartment, to generate continuously mature blood cells [1]. Recent studies have shown that the LMPP compartment is functionally plastic, dynamic, and more heterogeneous than expected [2]. LMPP retain lymphoid and myeloid potential, whereas the downstream common lymphoid progenitors (CLP) are mainly devoted generating the B cell lineage [3].

Following HSC transplantation, a delayed T cell reconstitution is observed compared to other lineages. The reasons are not yet elucidated. To overcome this difficulty, alternative approaches using downstream progenitors, such as LMPP or CLP, might be tested to accelerate lymphoid recovery. To reach this goal, it would require the development of new transplantation strategies for identifying the contribution of each progenitor to the development of the T cell compartment, an aspect that, for the moment, is not fully established.

Cellular barcoding methods have made it possible to differentiate the fate of cell populations that have very similar characteristics. The method is based on the insertion of a lentivirus (LV) carrying a given stretch of non-coding DNA, named barcode (BC), into a cell chromosome. Integrated BCs will replicate and segregate through cell divisions and, therefore, will be present in all cells’ progeny. The remarkable characteristic of the method is that different cell types can be tracked simultaneously in the same mouse to determine their outcome. Previous works using genetically barcoded HSC revealed unexpected characteristics [4,5,6]. Furthermore, other studies revealed a large heterogeneity in the proliferation and differentiation capacities of populations with similar phenotypes [7]. Therefore, this method is certainly the most appropriate strategy to quantify the dynamics of lymphoid progenitors moving towards the thymus.

Here, we report the first study comparing, in the same individual, the in vivo dynamics of the hematopoietic progenitors LMPP and CLP to T cells production by using a DNA barcoding strategy. The results highlight the predominant role of LMPP for lymphoid generation and reveal valuable novel insights to reconsider in clinical transplantation assays.

## 2. Results

### 2.1. Selection of Barcodes for Hematopoietic Progenitor Labeling

To analyze the LMPP/CLP dynamics, in particular to determine the differential contribution of these progenitors to lymphoid reconstitution, we have used a stretch of non-coding DNA, named barcode (BC), to track cells following competitive mouse transplantation assays.

Here, a set of 10 LV-encoding different BC were selected from our sequenced 50-BC collection. The BC are 150 bp long. They have identical sequences of 25 nt at the 5′- and 3′-end and are characterized by an internal sequence of 125 bp, unique for each BC (Figure 1A, Appendix A). The BC are located in the lentivirus, downstream of the gene expressing the eGFP, which is under the control of the Ubiquitin C promoter. This also allows for GFP tracking.

All the BCs can be amplified simultaneously by PCR using primers complementary to their identical sequences located at both ends of the BC (named TL and BL) (Figure 1A, Appendix A). Considering the differences in their internal sequences, it is essential to determine whether the different BCs can be amplified at the same rate so as not to produce biased results due to their differences in the GC percentage. For this purpose, we have designed specific primers at their unique internal sequence (named SpeB), which, in combination with a common forward primer complementary to a sequence located at the end of the GFP gene, named TB, can specifically amplify each BC (Figure 1A).

To test for the specificity of the designed primers and ensure that no cross-reaction might occur once located in the genome, we generated cells, each containing only one BC integrated into their genome. To this end, Lin**^-^** cells purified from CD45.1 B6 mice were transduced in vitro with each of the 10 BCs, in separate wells (one BC LV per well), in the presence of a cocktail of cytokines (Figure 1B). After 18 h, they were grafted independently into CD45.2 recipient mice. This gave rise to ten strains of mice, each possessing a unique BC in their genome.

The BM of the 10 different recipient mice was isolated after 6 weeks. The genomic DNA was extracted and amplified independently by PCR using the TB forward primer and each of the BC-specific reverse primers. Each primer was tested first with the corresponding DNA, and then, to look for cross amplifications, each primer was tested against all genomic DNA. Examples of BCs amplification with the TL/BL and TB//B are shown in Figure 2A. Their specific amplification with non-cross reactivity are shown in Figure 2B–D.

### 2.2. Barcoded HSC Transplantation for In Vivo LMPP and CLP Production

To obtain barcoded LMPP and CLP progenitors, HSC (Lin^-^c-Kit^+^Flt3^-^) issued from CD45.1 mice were transduced in vitro with 10 BCs independently. From these, two pools of HSC, each marked with 5 different BCs, were constituted (pool A and pool B). The HSC barcoded either with pool A or with pool B were then grafted independently to CD45.2 recipients (Figure 3A). We did two independent experiments, in which we modified the BC content of cells to avoid biased PCR amplifications due to differences in their GC content of their internal sequence.

The quality of the graft was first analyzed by determining the reconstitution of hematopoietic lineages in different tissues of CD45.2 recipient mice post-transplantation of the barcoded-HSC. We found that the percentages of CD45.1 cells in the BM, after 6 and 14 weeks, was 73.2% ± 5.0 (*n* = 6) and 62.1% ± 19.2 (*n* = 2); in the spleen, 83.9% ± 3.8 (*n* = 6) and 81.2% ± 0.3 (*n* = 2); and in the thymus, 98.1% ± 3.5 (*n* = 6) and 92.1% ± 6.3 (*n* = 2), respectively (Figure 4A). Successful reconstitution was also observed by measuring the presence of GFP^+^ cells in the three tissues (BM: range 33–46% GFP^+^ (mean = 45.6% ± 3.9); spleen: 30–46% (mean = 39.24% ± 1.8); thymus: 6–51%, mean = 27.6% ± 5.4) (Figure 4B). We then deepened our study by measuring the presence of CD45.1^+^GFP^+^ cells in myeloid (CD11b^+^) cells of the BM, in the B cells (CD19^+^) of spleen and BM, and the T cells (TCRβ^+^) in the spleen and the thymus (Figure 4C). Indeed, the T cell lineage was totally represented in the thymus (88.6 ± 4.4 of CD4^+^CD8^+^, DP) with a significant precursor pool (CD44/CD25), (Figure 4D) and the periphery (3.9 ± 3.7); the B cell compartment from the BM (17.32 ± 7.07) and the spleen (29.13 ± 4.9) was consistently present. Myeloid cells were detected in the BM (16.96 ± 6.32) as expected. Altogether, these results demonstrate the full capacity of barcoded-HSC to give rise to all hematopoietic lineages and further show the potential of this method to purify barcoded cells in vivo.

### 2.3. LMPP- and CLP-Barcoded Progenitor Production and Fate Determination

Given the successful reconstitution of hematopoiesis with the barcoded–HSC, we purified, from these transplanted mice, LMPP and CLP progenitors labeled either with BCs of group A or with BCs from group B by flow cytometry using specific markers (see Section 4). To analyze their outcome, two new groups were formed (Figure 3B). One was constituted of a mix of LMPP (500 cells) marked with BCs of pool A and 500 CLP cells marked with BCs of pool B. Inversely, the second group was constituted of a mix of LMPP (500 cells) marked with BCs of pool B and 500 CLP cells marked with BCs of pool A (Figure 3B). To increase the robustness of the results and prevent bias that can arise from different PCR amplification rates, we inverted the BC composition in the second experiment. Then, each group was grafted to 5 recipient mice separately as described in Section 4.

Following transplantation of barcoded progenitors, we first aimed to analyze the reconstitution capacities of these progenitors in recipient mice at early (6–9 weeks post-transplantation) and late (19–33 weeks) time points. The results show that the proportion of progenitor derived cells, in the BM, is reduced (Figure 5A). In the spleen, donor cells are reduced at the early time points, they accumulate to normal levels at latter time points. In contrast, the thymus is well repopulated at early times after graft of lymphoid progenitors. However, due to the low sustainability of these progenitors, together with their reduced self-renewal, late recipient mice showed reduced donor type cells (Figure 5A). The reduced thymopoiesis can be appreciated by the low proportions of DP (CD4^+^CD8^+^) cells and the Lin^-^ progenitor pool (Figure 5B,C).

Following transplantation of recipient co-grafted mice with LMPP and CLP progenitors, we observed in the BM maintenance of the myeloid lineage, although at a much lower proportion in comparison to recipient of HSC (Figure 6A). In contrast, in the B cell lineage, CD19^+^ cells accumulate in the BM, and their proportion increase with time, in mice co-grafted with LMPP and CLP progenitors (Figure 6B left panel). Similarly, in the spleen (Figure 6B right panel), B cells are present and their proportion is close to no manipulated mice. Mature T cells are present in the spleen (Figure 6C left panel) and reached normal proportion with time. Reduced proportions of DP in the thymus is observed, a consequence of the loss of self-renewal of the progenitors, unable to sustain the thymic repopulation (Figure 6C right panel).

Altogether, these data revealed that lymphoid B and T cell lineages are maintained over 33 weeks post-graft (Figure 6 and Appendix A for absolute numbers). Importantly, B cells become the dominant subset in the BM and significantly abundant in the periphery. These results reveal the good quality of the sorted progenitors as assessed by their profile of reconstitution, e.g., a reduced myelopoiesis to the advantage of lymphopoiesis.

### 2.4. Cell Distribution of Barcodes Highlights the Major Role of LMPP Progenitors

In order to analyze the origin of the lymphoid progeny in grafted mice, B (CD19^+^) and T (TCRβ^+^, CD4^+^, CD8^+^) populations were sorted and treated as described in Section 4.

Genomic DNA were then isolated from these populations, and PCR amplifications were performed to generate BC amplicons that were sequenced using Illumina MiSeq technology (1 × 150 bp and 1 × 300 bp). After read cleaning and alignment on BC genomic sequences, we selected only the samples having at least 10 k reads and 25% of uniquely mapped reads on BCs. A raw count of reads per BC and per sample was then obtained by summing the reads aligned for each BC-sample pair. Read counts per BC-sample range from 143 k to 408 k, with an average of 315 k (Appendix A).

A first normalization of the read counts was carried out to take into account the differences in sequencing depth between samples. A second normalization was then performed to eliminate the effect on counts of differences in amplification efficiency of each BC. This normalization was based on the control amplified with equal quantities of each BC. Finally, the normalized counts were converted into proportions to more easily compare and interpret the contribution of the BCs per sample. The proportion of BCs therefore specified each sample.

Fifteen mice were transplanted with the barcoded progenitors. In some cases, we made “mirror” mice, meaning that we inverted the BCs present in the LMPP compared to those contained in CLP cells. This avoids obtaining biased results due to internal sequence variations. To facilitate understanding, we show here the example of mouse 10, grafted with LMPP marked with BC1, BC4, BC6, BC12, BC25, and CLP with BC2, BC3, BC7, BC10, and BC15. Inversely, mouse 13 was grafted with LMPP/BC2, BC3, BC7, BC10, BC15, and CLP with BC1, BC4, BC6, BC12, and BC25. Consequently, the BCs present in LMPP in mouse 10 are present in CLP in mouse 13 and *vice versa*. Thus, mouse 10 can be considered as the “mirror” of mouse 13. The results of the analysis of the BC compositions of the progenitors are shown in Figure 7 (Appendix A). For both “mirror” mice (10 and 13), we found that the BC composition of BM, spleen, and thymus correspond to those of LMPP, with an abundance of 3 BCs LMPP/BC1, BC6, BC25 in mouse 10 and LMPP/BC2, BC7, BC15 in mouse 13.

The analysis of the same mouse performed at different times (Figure 8A) or, the comparison with other mice (Figure 8B), revealed similar profiles, which strongly adds to the accuracy of the results.

Notably, examination of the BCs within the B cell lineage in the BM and the spleen of transplanted mice revealed a broad spectrum of BCs. This is consistent with the fact that this cell population arose from the contribution of multiple progenitors of different origins. In contrast, the T cell lineage within the thymus is composed of only 1 or 2 BCs, suggesting that these are generated from a more limited diversity of LMPP progenitors.

Altogether, the results clearly demonstrate that cells in all subsets analyzed emerge from the LMPP, and therefore, it can be considered as the best candidate for the repopulation of lymphoid lineages in immunocompromised situations.

## 3. Discussion

This study compares for the first time the fate of the hematopoietic progenitors, LMPP and CLP simultaneously in the same mouse, with particular emphasis on T cell regeneration. The results highlight the predominant role of LMPP progenitors for lymphoid generation and the importance of focusing on this cell population to reconstitute immune-compromised organisms.

The outcome of these progenitors was highlighted using a cellular barcoding labeling method. The BCs integrated in the chromosome of cells by means of lentiviruses segregated through cell divisions and could be detected in all cells’ progeny. The remarkable characteristic of this strategy is that different cell types can be tracked simultaneously and in the same mouse. In particular, previous methods that might use congenic mice (CD45.1, CD45.2, Thy1.1, Thy1.2) or mice with a universal expression of fluorescent protein (GFP) need a large number of mice to produce similar and statistically strong results. Thus, an essential advantage of our sequencing-based cell labeling strategy, not found in other methods, is that it allows multiple cell populations to be followed simultaneously in the same mouse.

Based on these powerful properties, we could compare the capacities of LMPP and CLP populations to repopulate the thymus. The LMPP and CLP progenitors were barcoded in vivo to prevent in vitro steps that could alter their phenotype and fate, that is to say, circumvent the use of either irradiation or cytokines in cell cultures. Barcoded progenitors were then grafted to immunocompromised mice. The transplant experiments revealed that CLP are less efficient than LMPP for generating T cells. Many explanations can be raised, in particular that i) CLP might differentiate rapidly and generate mature cells earlier than LMPP and that ii) the kinetics to reach their specific niche might be different. In fact, separate niches might control LMPP and CLP since LMPP reside in HSC niches [8], whereas lymphoid progenitors CLP localize in endosteal niches [9,10,11].

Alternative, different pathways can be followed by LMPP and CLP for generating lymphoid cells, each depending on different regulatory factors. Consistently, it was reported that LMPP are more competent than CLP to achieve a full lymphopoiesis in the newborn situation [12]. Authors suggested that LMPP follow a CLP-independent stream, in comparison with the adult situation where LMPP generate an intermediate CLP stage before the production of mature lymphoid cells. In our conditions, we cannot exclude that this CLP-independent pathway is used by LMPP, since no CLP lymphoid reconstitution was observed. This alternative pathway for directly generating the lymphoid lineage from LMPP might be set up following stress conditions (immune-compromised mice).

Our studies also reveal that progenitors can be maintained for long periods in transplanted mice retaining their full competence for lymphoid regeneration. This makes them good candidates for lymphopenic situations and/or auto-immunity afflictions.

To increase the value of our results on the fate of progenitors, we performed a double check experiment. LMPP cells were labeled with five different BCs and grafted together with CLP possessing five different ones. Then, in a subsequent experiment, the BCs were inverted to produce “mirror mice” as explained in the Results section. This prevents bias related to different amplification rates due to the differences in the GC percentage of the BC internal sequence. The barcodes in the different cell types were carefully quantified by adjusting several parameters and filters such as the percentage of reads per sample, sequencing depth and verification of uniquely mapped read on single reference BC. The results obtained with the “mirror” mice reflect the solidity of our strategy and effectively show the dominance of LMPP in lymphoid reconstitution.

Notably, sequencing the BCs present within B cell subsets (CD19^+^) in the BM and spleen revealed that multiple BCs are present. This suggests that multiple clones participate to the reconstitution of these cells. In contrast, sequencing the T cell lineage, thymic subset, and peripheral T cells revealed the presence of a limited number of BCs. This is consistent with the contribution of very few clones to thymus repopulation. Our data recall the report of FJT Staal’s laboratory showing that only few HSC clones will contribute to T cell development. They report that despite the restricted number of HSC clones entering the thymus, a full mature T cell receptor repertoire can be generated [13]. Thus, intrathymic events might control progenitors’ entry.

Overall, our findings highlight the role of progenitors in hematopoietic reconstitution and might provide the basis for novel strategies for their use in clinical settings.

## 4. Materials and Methods

### 4.1. Mice

C57BL/6 CD45.1 Thy1.2 (B6) and C57BL/6 CD45.2 Thy1.1 (BA) mice were used at 10–18 weeks-old (both males and females). They were purchased from CDTA (Orléans, France) and were kept in specific pathogen-free facility of SFR Necker (Agreement n°75–1026). All experimental procedures using animals were approved by the “Comité d’éthique en expérimentation animale’’ of the Université Paris Cité and the French Ministry of Research, Innovation, and Education (APAFIS #26599-2019071812345604).

### 4.2. Antibodies and Cytometer

The following fluorescent conjugated antibodies were used for staining and cell sorting and were obtained from BD Bioscience (Le Pont de Claix, France), Thermo Fisher Scientific (Villebon, France), or SONY (Weybridge, United Kingdom): anti-NK1.1 (PK136), anti-TCRβ (H57-597), anti-CD3ε (145-2C11), anti-Mac 1/CD11b (M1/70), anti-CD19 (1D3), anti-GR 1 (RB6-8C5), anti-Ter 119 (Ly76), anti-Flt3/CD135 (Flk2; A2F 10.1), anti-c-Kit/CD117 (SCF receptor; 2B8), anti-Sca 1 (E13-161.7), anti-IL7Rα/CD127 (A7R34), anti-VCAM 1/CD106 (429), anti-CD45.1 (A20), anti-CD45.2 (104), anti-CD4 (GK1.5), and anti-CD8 (SK1). They were directly coupled to FITC, APC, APC-eF780, PE, PerCP-Cy5.5, PE Cy7, and APC Cy7 or conjugated with biotin, the latter being revealed by streptavidin-PE-Cy7. All staining was performed at 20 min at 4 °C with agitation.

Flow cytometry data were acquired using FACS Canto II running the DIVA software v9.0.1 (BD Biosciences). Cell sorting was performed using FACS Aria III running the DIVA software. Flow cytometry data analysis was performed using the FlowJo software v10.8.1 (BD Biosciences).

### 4.3. Isolation of Bone Marrow Progenitors

Bone marrow cells from B6 mice were collected by crushing femurs, tibia, hips and spine bones. Then cells were incubated with unconjugated rat antibodies against B220, Ter119, GR1 and Mac1 (which are specific for B cells, erythroid cells, dendritic cells and macrophages, respectively). Positive cells were magnetically depleted with sheep anti-rat IgG-conjugated beads and sheep anti-mouse IgG-conjugated beads (Dynabeads M-450; Dynal Thermo Fisher Scientific, Villebon, France). The remaining cells (Lineage-negative cells, Lin- enriched) were labeled with fluorescent conjugated antibodies against c-Kit, Sca-1 and lineage (Lin) Ags (NK1.1, TCRβ, CD3ε, Mac1, CD19, GR1, and Ter119).

The LMPP population was sorted by the phenotype of Lin^-^ Sca-1^+^c-Kit^+^(LSK)Flt3^+^. VCAM-1 antibody was used to identify two subsets of LMPP: MPP2 (LSK Flt3^+^VCAM-1^+^) and MPP3 (LSK Flt3^+^VCAM-1^−^). The CLP population was sorted by the addition of anti-IL7Rα and their phenotype was Lin^-^ Sca-1^lo^c-Kit^lo^IL7Rα^+^ (Appendix A).

### 4.4. Cells Transduction

Sorted HSC (LSK Flt3^−^) cells were pre-activated overnight in StemPro-34 SFM medium (Thermos Fisher Scientific, Villebon, France) completed with 1% glutamine, 15% FBS, and 1% antibiotics and cytokines: SCF 100 ng/mL, Flt3L 100 ng/mL, TPO 20 ng/mL, and IL6 20 ng/mL. Then, the cells were infected with barcoded lentiviral vector at a multiplicity of infection (MOI) of 50 per cell in StemPro-34 SFM medium completed with 1% glutamine, 15% FBS, and 1% antibiotics and cytokines: SCF 100 ng/mL, Flt3L 10 ng/mL, IL3 10 ng/mL, and IL6 20 ng/mL. After 15 h of infection, cells were washed with StemPro-34 SFM medium without antibiotics or cytokines. All cytokines were of murine origin (ImmunoTools, Friesoythe, Germany). The percentage of transduced cells (GFP^+^) is analyzed 72 h after the end of infection by flow cytometer.

### 4.5. Grafting Strategy

Two groups of barcoded LSK were constituted (Appendix A) in PBS with 2% FBS and grafted i.v. into sublethally irradiated (700 Rad) female BA as 1st recipient mice. After 6 or 9 weeks, they were sacrificed to harvest BM for sorting progenitor cells. An amount of 500 CLP GFP^+^ and 500 LMPP GFP^+^ from the two groups of each primary recipient were grafted with 10^5^ helper BM cells from IL7Rα KO mice (CD45.2^+^) into sublethally irradiated (700 Rad) congenic BA mice (CD45.2^+^) as second recipient mice. At 6, 7, 8, 13, and 33 weeks after transplantation, secondary recipient were sacrificed to harvest donor mature cells from BM, spleen and thymus cells defined by specific mature markers: B cells (CD19^+^ CD45.1^+^), NK cells (NK1.1^+^ CD45.1^+^), Myeloid cells (CD11b^+^ CD45.1^+^) and T cells (TCR^+^ CD45.1^+^). In the thymus, CD4, CD8, CD44, and CD25 were included to sort Simple Positive cells (SP): CD4^+^ or CD8^+^, Double Positive cells (DP): CD4^+^ CD8^+^ and Triple Negative cells (TN): CD4^−^ CD8^−^ CD3ε^−^. Mean and SEM of recovered cells were calculated using GraphPad Prism software v.9.4.1.

### 4.6. Genomic DNA Extraction for Barcode Analysis

After sorting, cells were washed in PBS 1X, centrifuge 5 min, 4 °C at 1000 g. The dry pellets were frozen in liquid nitrogen and then kept at −80 °C. Genomic DNA extraction was performed with the kit Nucleospin Genomic DNA from Tissue mini (Macherey-Nagel, Germany) according to the manufacturer protocol and quantified using the DNA quantification Kit from Bio-Rad (Marne-La-Coquette, France).

### 4.7. Analysis of Barcoded Genomic DNA

PCR-amplification of barcoded genomic DNA was initiated with 40 ng of genomic DNA under the following conditions: 95 °C for 5 min; 40 cycles: 95 °C for 10 s, 65 °C for 30 s, and 72 °C for 50 s. All BCs were amplified using primers surrounding the BCs: TL (TGCTGCCGTCAACTAGAACAC) in combination with BL (GATCTCGAATCAGGCGCTTA) or the TB (GTCCTGCTGGAGTTCGTGAC) coupled with BB (GCCATACGGGAAGCAATAGC). The location is schematized in Figure 1A. To amplify specifically each BC, we used primer CAAAGACCCCAACGAGAAGC, located at the end of the GFP gene and a specific primer complementary to the internal and specific BC sequence (Appendix A).

The design of specific 3′ primer of 21–24 bp in length was performed for each BC using Primer3 web tool [9]. Alignment stringency and specificity has been checked by local alignment (BLAST) to others BCs sequences [10].

To avoid bias in the quantification of amplified fragments, we took into account the PCR bias that might be introduced by the GC percentage differences for each BC. Therefore, we normalized results with the PCR yield for each BC. This was calculated using an equimolar pool of genomic DNA prepared from cells barcoded with independent BCs and a pool of genomic DNA prepared from individually barcoded cells at a ratio that exactly reproduces the BC composition established in vitro.

### 4.8. Sequencing of Tissue Extracted Barcodes

The amplified fragments were further extended with specific multiplex identifier (MID) sequences. Fourteen different MID sequences were used, allowing for pooling and simultaneous sequencing of 14 different samples. The PCR was initiated with 40 ng of genomic DNA under the following conditions: 95 °C for 5 min; 40 cycles: 95 °C for 10 s, 62.3 °C for 30 s, and 72 °C for 50 s. This first PCR generates the same three sizes of amplicon quantified with the ImageJ software. Considering the limited size of the BC repertoire, Illumina MiSeq sequencing of PCR fragments amplified from genomic DNA of different sorted cell populations is used to robustly and sensitively identify and quantify the BC repertoire. Different pools, composed of the same quantity of genomic DNA derived from cells labeled with only one BC, were sequenced to calculate the overall method yield for each BC.

### 4.9. Bioinformatics Processing of BC Reads

The quality of Illumina MiSeq reads (1 × 150 bp or 1 × 300 bp) generated from the DNA-sequencing of each cell-sorted population was assessed using CNRGH in-house bioinformatics quality control pipeline and FastQC [14]. After BC reference and index creation, single-end reads were trimmed with Trimmomatic software (HEADCROP:26 CROP:98 MINLEN:36) to keep only the central part of the read corresponding to BC specific sequence [15]. Reads from each sample were aligned on the BCs genomics reference sequences by bowtie2 mapper using its end-to-end mode configured with very sensitive parameters for higher accuracy (-p 8 -D 20, -R 3, -N 1, -L 10, -i S,1,0.50) [16]. Only reads uniquely mapped on single BC and over 95 bp of their length were conserved. Raw count table of the aligned reads per BC and per cell population was then generated. Samples have been selected according to filters on sequencing depth, with at least 10 k reads per sample, and 25% of reads aligned one time on BCs reference sequences.

### 4.10. Normalization of BC Read Counts

To normalise sequencing depth between samples, for each BC, raw counts were divided by the total sum per sample and multiplied by 1000, leading to a normalized count per BC in CPK (Counts Per thousand reads). To normalise PCR amplification differences between BC, a reference sample containing each BC in equimolar proportion was used. BC in CPK for each sample was divided by the corresponding BC in CPK from the reference sample and multiplied by 100 (Appendix A).

## Figures and Tables

**Figure 1 ijms-24-04368-f001:**
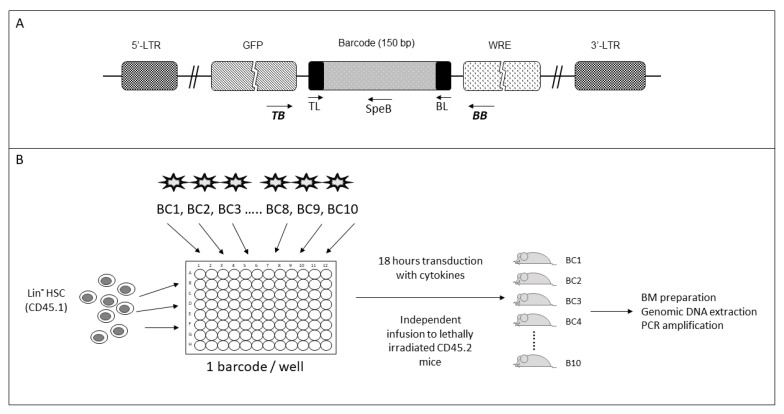
Schematic representation of the barcode structure (**A**) and the strategy to track barcoded cells in mice (**B**). (**A**) Diagram of the lentiviral vector carrying a 150 bp BC which is located downstream of the eGFP gene. The location of primers used to amplify all the BCs simultaneously (TL/BL or TB/BB) or each BC individually (TB/SpeB) are shown. (**B**) The BC lentiviral vector strategy used to transduce CD45.1 HSC of Lin^−^c-Kit^+^FLT3^−^ phenotype. One BC per well. Following 18 h, transduced cells were infused into CD45.2 myelo ablated mice, and the BCs analyzed in different tissues of transplanted mice at different times.

**Figure 2 ijms-24-04368-f002:**
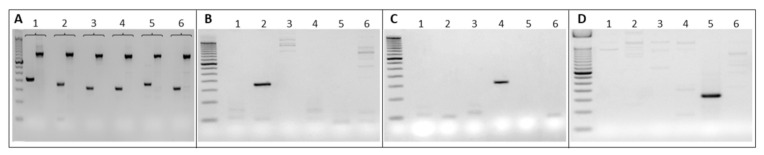
Specificity of BCs. Examples of gel electrophoresis showing the specificity and the absence of cross-reaction following BC amplification by PCR with the specific primers. (**A**) Example of BCs amplified with the common TBB/SpeB primers (specific band around 150 bp) and the TB/BB primers (550 bp). (**B**–**D**): examples of the specificity of the designed primers. Six genomic DNAs (1 to 6) were amplified with primers 2, 4 and 5 respectively. The figure shows that primer 2 amplifies only DNA 2 (**B**); primer 4, amplifies only DNA 4 (**C**) and primer 5, amplifies only DNA 5 (**D**), consistent with the absence of cross-reaction.

**Figure 3 ijms-24-04368-f003:**
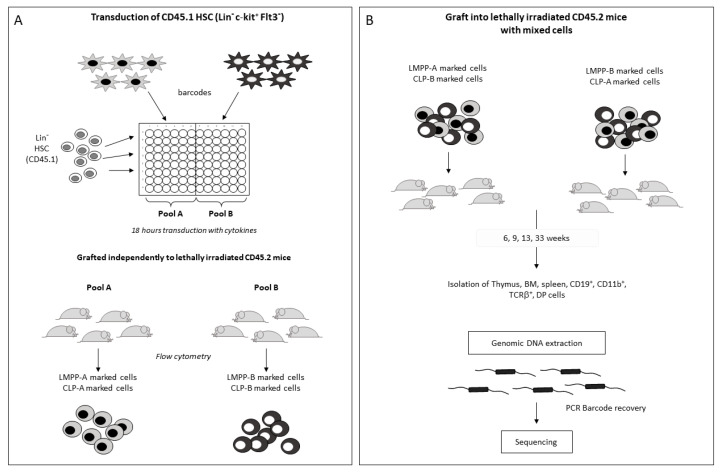
Strategy to obtain barcoded LMPP and CLP populations in vivo. (**A**) Lin^−^c-Kit^+^Flt3^−^ CD45.1 HSC were marked independently with 10 different BCs. From these, two pools of cells (pool A and pool B) were constituted. Each pool is a mix of cells containing 5 different BCs. Both pools were grafted into CD45.2 mice. BM cells of the transplanted mice were then prepared and sorted with specific markers to obtain barcoded LMPP and CLP. (**B**) To determine the fate of each progenitor, two groups were further constituted. Group 1 was composed of equal numbers (500 cells) of LMPP-barcoded with pool A + CLP-barcoded with pool B, and group 2 was constituted of LMPP-barcoded with pool B + CLP-barcoded with pool A. These were grafted into lethally irradiated CD45.2 mice. Genomic DNA extraction was performed from the different tissues at different time points and amplified by PCR with the specific primers shown in (Figure 1A). Sequencing was then performed as described in Section 4.

**Figure 4 ijms-24-04368-f004:**
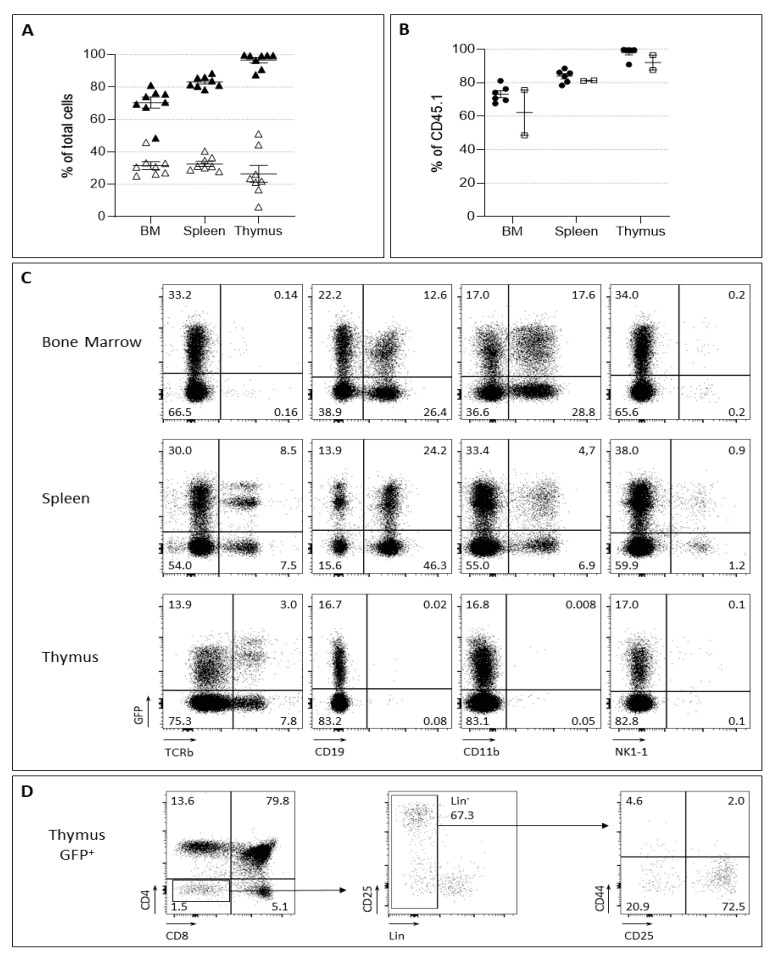
Analysis of the barcoded HSC reconstitution potential. (**A**) Percentages of donor type cells CD45.1^+^ in the bone marrow (BM), spleen and thymus (dark triangles). The percentages of CD45.1+GFP^+^ cells is also indicated (empty triangles). Each dot represents one mouse. (**B**) Mice were studied 6 weeks post-graft (dark dots) or 14 weeks post-graft (empty square dots) with barcoded HSC. The percentage of CD45.1^+^ donor type cells detected in the BM, spleen and thymus is indicated for all mice analyzed. (**C**) An example of barcoded HSC grafted mice analyzed 6 weeks after transplantation. BM, thymus and spleen were studied for the presence of myeloid (CD11b^+^), B (CD19^+^), NK (NK1.1 and T (TCRβ)) populations. (**D**) Analysis of the thymus of the same mouse. Gating within the donor CD45.1^+^ GFP^+^ is shown. Mature thymic subsets (CD4, CD8) as well as progenitors (Lin^-^CD44/CD25) are present.

**Figure 5 ijms-24-04368-f005:**
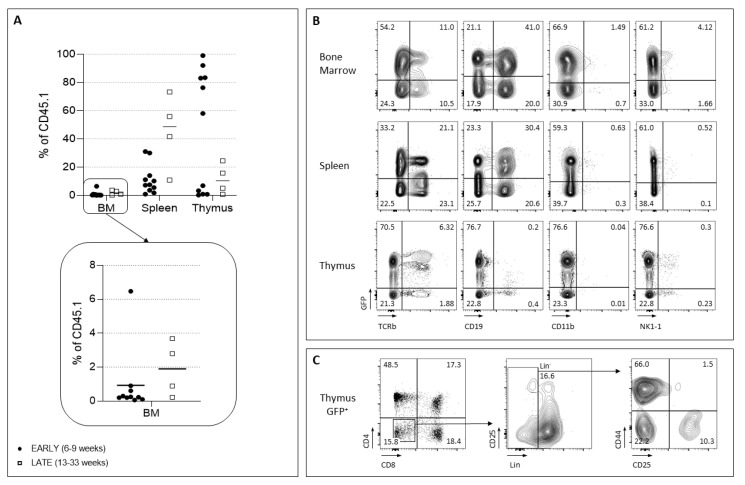
Analysis of the barcoded LMPP and CLP reconstitution potential. (**A**) Mice were studied at early (6–9 weeks) or late (13 weeks) time points post-graft with BCs LMPP-and CLP-transduced in vivo. The percentage of CD45.1^+^ donor type cells detected in the BM, spleen and thymus is indicated for all mice analyzed. Each dot represents one mouse. A highlight of the values in the BM is shown. (**B**) An example of LMPP/CLP grafted mice analyzed 9 weeks later. BM, thymus and spleen were studied for the presence of myeloid (CD11b^+^), B (CD19^+^), NK (NK1.1) and T (TCRβ) populations. (**C**) Analysis of the thymus of the same mouse. Gating within the donor CD45.1^+^ GFP^+^ is shown. Mature thymic subsets (CD4, CD8) as well as progenitors (Lin^−^ CD44/CD25) are present.

**Figure 6 ijms-24-04368-f006:**
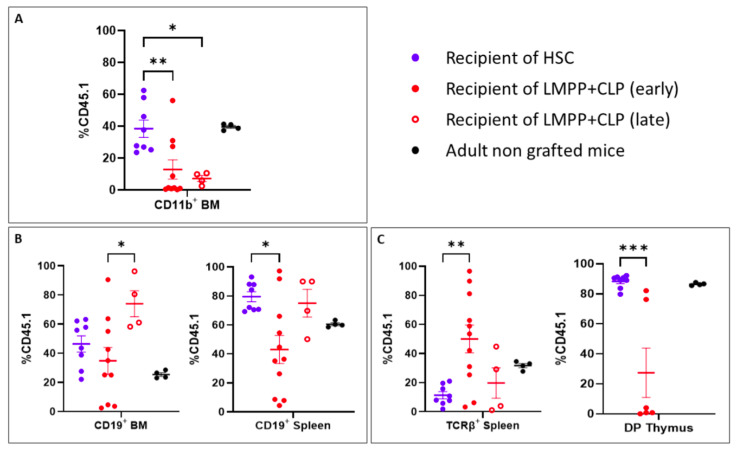
Comparison of the reconstitution potential of HSC and LMPP/CLP. Evaluation of the myeloid lineage CD11b^+^ in the BM (**A**), B cell lineage in the BM and spleen (**B**) and T cell lineage in the spleen and thymus (**C**) of mice grafted with HSC (open black dots), or LMPP/CLP progenitors (closed black triangle dots for early time point and open black triangle dots for late time points). Black dots represent the indicated lineages at the steady state in normal unmanipulated mice. Each dot represents one mouse. Bars represent mean ± SEM, * *p* < 0.05, ** *p* < 0.01, *** *p* < 0.001.

**Figure 7 ijms-24-04368-f007:**
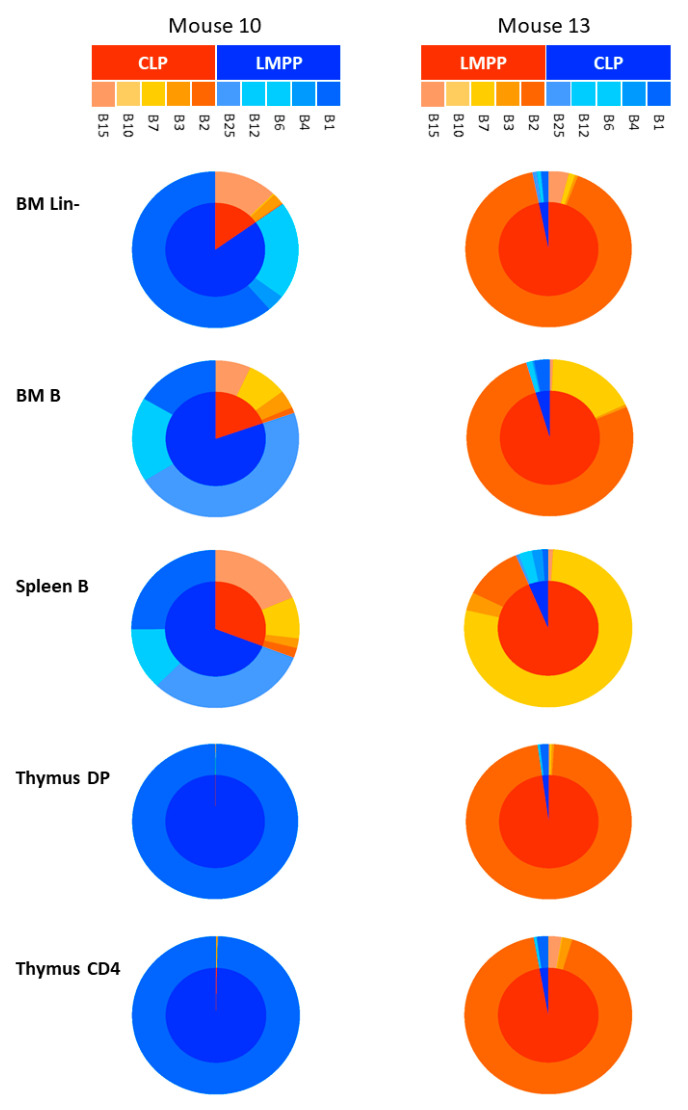
Sunburst plots representing the contribution of LMPP/CLP progenitors to the reconstitution of lymphoid populations. The BC composition of BM, spleen and thymus of two representative “mirror” mice (10 and 13) is shown. Each BC is represented by a color code. Orange to red BCs are present in CLP of mouse 10 and in LMPP of mouse 13. The blue BCs are present in LMPP of mouse 10 and in CLP of mouse 13. The inner circles represent the dominant BC color within a mentioned tissue. The blue color in mouse 10 is dominant and associated with LMPP subset. In contrast, the red color is dominant in mouse 13 and associated also with the LMPP subset. The outer circle details individual BCs found in each cell type.

**Figure 8 ijms-24-04368-f008:**
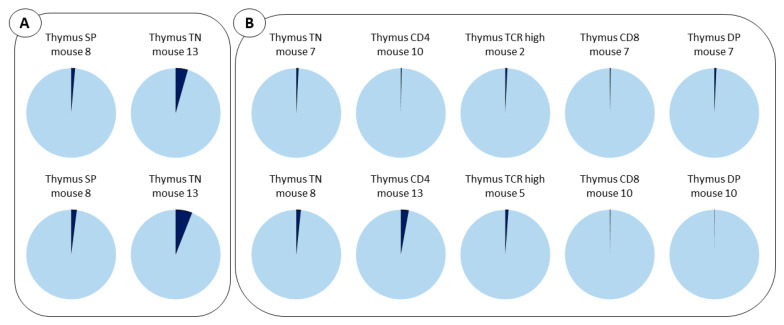
Barcodes analysis in multiple mice and subsets. (**A**). Results of thymus SP (CD4^+^ or CD8^+^) and thymus TN (CD3^−^CD4^−^CD8^−^) obtained for the same mouse in two independent experiments. (**B**). BC composition of thymus N, CD4, TCR high, CD8 and DP (CD4^+^CD8^+^) analyzed in different mice. LMPP BCs are shown in light blue and CLP in dark blue.

## Data Availability

The DNAseq data sets can be accessed through ArrayExpress (https://www.ebi.ac.uk/arrayexpress; accession number E-MTAB-12745).

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
