# Peer review of "Co-Transplantation of Barcoded Lymphoid-Primed Multipotent (LMPP) and Common Lymphocyte (CLP) Progenitors Reveals a Major Contribution of LMPP to the Lymphoid Lineage"

_ijms, 2023, doi:10.3390/ijms24054368_

Round 1
Reviewer 1 Report
In the study, the authors, utilizing a lentiviral infection conducted barcode labeling technology, conclude that LMPPs show a higher potential in generating lymphoid cells. Though the results are clearly presented, the design of the experiments and the conclusion seem to lack of necessity and novelty.
1. Barcode labeling is a powerful tool for cell fate tracking. However, in this study, only limited number of barcodes are utilized. An alternative strategy, may be much easier, is to just use congenic mice (CD45.1, CD45.2, Thy1.1, Thy1.2) or mice with an universal expression of fluorescent protein (GFP) to get the combination of progenitor cells for transfer. The result can be directly checked by flow cytometry analysis, without any potential variation as may happen during PCR and sequencing. The author need to find a way to emphasize the advantage of their technology in solving the problem.
2. LMPP are in an earlier developmental stage than CLP. It is not surprising that they show a higher potential in generating lymphoid cells. Did the authors also checked the generation of CLP by the barcode labeled LMPP? Do they also continuously generate more CLP than a single CLP transfer? What is the ratio between LMPP generated CLP and the transferred CLP in BM? Does the ratio correlates with the ratio between lymphocytes in periphery?
Reviewer 2 Report
The paper proposed by Lopez et al. is extremely interesting and novelty, the first one to address the in vivo dynamics of the hematopoietic progenitors LMPP and CLP to T cells production by using a DNA barcoding strategy, while also pointing to new research directions regarding the role of LMPP in correlation with transplantation.
I would like to congratulate the authors for the methodology of the study and the presentation of the results.
The methodology has been comprehensively described. Figures and explanations are appropriate to the content of the manuscript.
Minor linguistic changes are necessary.
Round 2
Reviewer 1 Report
1. I totally agree that the barcode labeling is a new strategy and may be a powerful tool for solving an appropriate problem. However, its advantage in this study is not so obvious. The authors explain that their barcode labeling strategy allows reduced number of mice used in an experiment. This may be a good point, but is not necessarily to be always the case, depending on how many progenitor cells are needed to transfer to sufficient recipient mice. In addition, the barcode labeling takes an additional in vitro infection procedure, which not only may cause cell loss, but also could change the status of the progenitors.
And, I actually don't understand why CD45.1, CD45.2, Thy1.1, Thy1.2 markers and GFP expressing mice cannot be used to track them within a recipient mouse simultaneously. First, through breeding, they can generate as many as 18 strains, which is more than the number of barcodes used in this study. Secondly, the progenitors can be purified from each of these strains, and mixed together to transfer into a same recipient mouse, which will enable the tracking of them simultaneously.
This study also require flow cytometric analysis and cell sorting. CD45.1, CD45.2, Thy1.1 and Thy1.2 detection by flow cytometry is quite easy and reliable. Unlike PCR, it does not need cell sorting. So, it is easier to track the differentiation of the progenitors to multiple lymphoid or myeloid lineages with additional antibodies added to the staining. In contrast, though PCR are sensitive, it also generates some false positive results. And, after PCR amplification, the percentage of progeny cells generated by each barcoded progenitors is more likely to be affected.
2. The explanation for the next point is actually what I concerned. Given CLP is a transient progenitor population, without self-renewal, the significant of this study is reduced. The quantification results may largely relay on after how long of the transfer the authors analyzed the recipient mice.
Author Response
"Please see the attachment."
